# Effect of β-caryophyllene from Cloves Extract on *Helicobacter pylori* Eradication in Mouse Model

**DOI:** 10.3390/nu12041000

**Published:** 2020-04-04

**Authors:** Da Hyun Jung, Mi Hee Park, Chul Jin Kim, Jin Young Lee, Chang Yeop Keum, In Seon Kim, Chang-Hyun Yun, Sung-kyu Kim, Won Ho Kim, Yong Chan Lee

**Affiliations:** 1Department of Internal Medicine, Yonsei University College of Medicine, Seoul 03722, Korea; 2SFC bio Co., Ltd., 119, Dandae-ro, Dongnam-gu, Cheonan-si, Chungnam 31116, Korea; 3New Drug & Bio Research Center, Handok Inc. 2nd floor, A-dong, BioPark, 700, Daewangpangyo-ro, Bundang-gu, Seongnam-si 13488, Korea

**Keywords:** β-caryophyllene, *Helicobacter pylori*, eradication

## Abstract

New antibacterial treatments against *Helicobacter pylori* are needed as *H. pylori* is acquiring antibiotic resistance. β-caryophyllene is a natural bicyclic sesquiterpene, with anti-inflammatory and antimicrobial effects. This study investigates the effects of H-002119-00-001 from β-caryophyllene on the eradication of *H. pylori* in a mouse model, and its effects on the inflammation of the gastric mucosa. To evaluate the anti-*H.pylori* efficacy of β-caryophyllene, a total of 160 mice were divided into eight groups (*n* = 10 each) and were administered different treatments for 2 and 4 weeks. *H. pylori* eradication was assessed using a Campylobacter-like organism (CLO) test and *H. pylori* qPCR of the gastric mucosa. The levels of inflammation of gastric mucosa were assessed using histology and immunostaining. H-002119-00-001 decreased bacterial burden in vitro. When H-002119-00-001 was administered to mice once daily for 2 weeks, cure rates shown by the CLO test were 40.0%, 60.0%, and 70.0% in groups 6, 7, and 8, respectively. *H. pylori* levels in gastric mucosa decreased dose-dependently after H-002119-00-001 treatment. H-002119-00-001 also reduced levels of inflammation in gastric mucosa. H-002119-00-001 improved inflammation and decreased bacterial burden in *H. pylori*-infected mouse models. H-002119-00-001 is a promising and effective therapeutic agent for the treatment of *H. pylori* infection.

## 1. Introduction 

*Helicobacter pylori* is a Gram-negative spiral-shaped bacillus that infects more than half of the global population [1,2]. Generally, *H. pylori* infection is acquired during childhood, and if left untreated, it can remain with the body of the patient throughout his/her life [3,4]. *H. pylori* colonization causes a number of human diseases, including chronic gastritis, peptic ulcer, gastric adenocarcinoma, and primary gastric lymphoma [5,6]. Therefore, eradication is important for treating and preventing *H. pylori* infection-related diseases [7,8]. Standard triple therapy (STT), which consists of proton pump inhibitor (PPI), amoxicillin, clarithromycin or metronidazole, was developed in the 1990s and is recommended as the first-line eradication therapy due to a substantial *H. pylori* eradication rate [9,10]. However, the rate of *H. pylori* eradication by STT has decreased because of increasing antibiotic resistance. Therefore, there is an increasing demand for safe and effective non-antibiotic compounds that inhibit the growth of *H. pylori* [11,12,13]. Recent studies have shown that lower incidence of *H. pylori* infection has been associated with the consumption of many foods of vegetal origin, including green tea and wine, which are rich in phytochemicals such as anthocyanidins, tannins, flavones, isoflavones, flavo- and flavanols, and stilbene derivatives [14,15,16]. β-caryophyllene is a natural bicyclic sesquiterpene that is present in a wide range of plant species such as cloves, basil, and cinnamon leaves, and copaiba balsam. [17] It ((1R,4E,9S)-4,11,11-trimethyl-8-methy lidenebicyclo [7.2.0] undec-4-ene) is a major constituent of many essential oils obtained from a number of plant species such as the Syzygium (~13%), Betula (~30%), and Strobilanthes (~7%) species [18,19,20]. It has been known to have anti-inflammatory, antioxidant, anticancer, and antimicrobial effects [21,22,23,24]. In this study, we used the β-caryophyllene from cloves, which are the aromatic flower buds of a tree, Syzygium aromaticum, in the family Myrtaceae. For the preparation of β -caryophyllene, steam distillation of dried flower’s buds was performed and then the essential oil was obtained. After fractional distillation and concentration of essential oil, β-caryophyllene was prepared. H-002119-00-001 is obtained from β-caryophyllene extract. In this study, we investigate the effects of the non-antibiotic compound H-002119-00-001 on the eradication of *H. pylori* in a mouse model and study its effect on the inflammation of gastric mucosa. 

## 2. Materials and Methods

### 2.1. Materials

H-002119-00-001 was obtained from SFC Bio., Cheonan, Korea. Metronidazole (MTN), clarithromycin (CLR), and omeprazole were purchased from Sigma-Aldrich Corp. (St Louis, MI, USA). 

### 2.2. Ethics

The Institutional Animal Care and Use Committee at the National Center of Efficacy Evaluation for the Development of Health Products Targeting Digestive Disorders, Incheon, Korea, approved the animal procedures conducted on rats.

### 2.3. In Vitro Quantification of Colony Forming Units (CFUs)

*H. pylori* KCTC12083 (Korea Collection for Type Culture, Daejeon, Korea) was used for the in vitro experiment. Bacteria were cultured in Columbia agar with 5% sheep’s blood and incubated at 37 °C under micro-aerobic conditions. To determine the antimicrobial effect of β-caryophyllene, *H. pylori* KCTC12083 was inoculated into (10^5–6^ CFU/mL) 10 mL phosphate-buffered saline (PBS) and dilutions of 1:10, 1:100, and 1:1000 H-002119-00-001 were prepared. H-002119-00-001 was added to the suspension of *H. pylori* KCTC12083 and vortexed. The mixture was inoculated onto Columbia agar plates containing sheep’s blood (5%). The no-treatment group served as the control group. The number of bacterial colonies was determined after 84 h.

### 2.4. Inoculation of Experimental Animals

Male C57BL/6 mice (4-week-old) were purchased from Central Lab. Animal Inc., Seoul, Korea. *H. pylori* SS1 strain was used for inoculation. Bacteria were maintained in trypticase soy agar (Difco Laboratories Inc., Detroit, MI, USA) containing sheep’s blood (5%) and incubated at 37 °C under micro-aerobic conditions (10% CO_2_, 85% N_2_, and 5% O_2_) for 2–3 days. For the assessment of anti-*H. pylori* effect in vivo, 160 mice were acclimatized for 1 week before the experiment. After acclimatization, mice were fasted for 12 h before *H. pylori* infection. Towards this, 120 mice were orally administered 200 μL of 5.0 × 10^9^ CFU/mL *H. pylori* suspension for three times at 2-day intervals. Non-infected mice, serving as the normal control group, were orally administered an equivalent volume of PBS. 

### 2.5. Distribution of Animals

After 2 weeks of *H. pylori* infection, plasma was isolated from the facial vein blood from all mice. Antibodies against *H. pylori* were measured using the mouse *H. pylori* antibody (IgG) ELISA kit (Cusabio Biotech Co., Houston, TX, USA), and only those mice that exhibited elevated *H. pylori* IgG levels were used for further experiments. After this, the mice were divided into eight groups (*n* = 10): Group 1 and Group 3 were orally administered with 5 mL/kg corn oil every day for 2 weeks; Group 2 and Group 4 were orally administered 5 mL/kg of 0.5% carboxymethyl cellulose sodium salt (CMC) every day for 2 weeks; Group 5 was orally administered with 5 mL/kg of 14.2 mg/kg metronidazole (MTN) + 7.15 mg/kg clarithromycin (CLR) + 138 mg/kg omeprazole every day for 2 weeks; Group 6 was orally administered with 5 mL/kg of 100 mg/kg H-002119-00-001; Group 7 was orally administered with 5 mL/kg of 200 mg/kg H-002119-00-001; Group 8 was orally administered with 5 mL/kg of 500 mg/kg H-002119-00-001. H-002119-00-001 was administered to the mice once daily for 2 and 4 weeks (Figure 1A,B). 

### 2.6. Bacterial Identification 

At the end of the experiment, animals were euthanized, and stomachs were removed from their abdominal cavities. The gastric mucosa from the pylorus was biopsied for the Campylobacter-like organisms (CLO) test.

### 2.7. CLO Test

The gastric mucosal tissue extracted on the day of autopsy was aseptically collected and tested using CLO test reagent (Asan Pharmaceutical Co., Seoul, Korea). After 2 h of incubation at 37 °C, if the resultant color changed from yellow to red, the result was determined to be positive. After the CLO test, the mean and standard deviation for each group were determined by using the following scale: 0 points for no color change in the medium, 1 point for slightly red color, 2 points for light purple, and 3 points for dark purple.

### 2.8. Real-Time Polymerase Chain Reaction (qpcr) for Identifying H. Pylori in Gastric Mucosa 

RNA was collected from gastric mucosal tissue, and DNA was synthesized to perform PCR. qPCR was performed using specific primers. The marker gene used in this experiment was the 16S rRNA specific for *H. pylori*. Primers specific for *H. pylori* 16S rRNA were used; these were 5’-CTTAACCATAGAACTGCATTTGAAACTAC-3’ (forward) and 5’-GGTCGCCTTCGCAATGAGTA-3’ (reverse). The results represent the mean and error between groups through the relative quantitative calculation of the ΔCt (threshold cycle) value.

### 2.9. Histopathological Assessment 

#### 2.9.1. Histological Evaluation

Stomachs were removed and dissected along the greater curvature in the different mice groups. These were then fixed in 10% formalin, paraffin-embedded, sectioned to 4 μm, and stained with hematoxylin & eosin (H&E), and histopathological examination was performed. Histopathological scores indicated overall gastritis with damage to the surface epithelium, inflammatory cell infiltration, and submucosal edema, all of which were collectively found in the entire corpus and antrum region of mice. After measuring the degree of atrophic gastritis using the strategy used in previous studies [25,26], the grade of each tissue was recorded in terms of scores. Then, the sum of the detailed scores of each organization was calculated, followed by measuring the average score of each group; the scores were calculated by two different researchers to increase objectivity. The score was classified as 0 points if not observed for each item, 0.5 points for mild, and 1 point for moderate.

#### 2.9.2. Immunohistochemistry

After paraffin embedding, tissue slides were subjected to immunostaining through deparaffinization and hydration. Antigen retrieval was performed to help in visualizing the antigen after blocking the endogenous enzyme in the tissue (blocking of endogenous enzyme). After staining with a macrophage-specific F4/80 antibody (Monoclonal Antibody (BM8), #14-4801-82, Thermo Fisher), the sample was probed with the secondary antibody, followed by DAB treatment. After the nuclei were stained with hematoxylin, samples were dehydrated, mounted, and observed under a microscope. Referring to a previous study [27], mild and moderate can be expressed as follows: moderate (1+) in case of positive reaction in both lamina propria and muscularis mucosa in gastric tissue, none (0) in otherwise, and mild (0.5+) in the meantime. The average score of each group was calculated, and scores were calculated based on observations by two different researchers to increase objectivity.

### 2.10. Statistical Analysis

Data are presented as means ± standard error (SE), and groups were compared using the non-parametric Mann–Whitney test. A *p*-value < 0.05 was considered statistically significant. The results were statistically processed using Sigma plots (Sigmaplot 12.2, Systat Software Inc., San Jose, CA, USA).

## 3. Results

### 3.1. In Vitro Effect of H-002119-00-001 on Bacterial Colonization

The antimicrobial efficacy of H-002119-00-001 against *H. pylori* KCTC12083 was evaluated at a dose of 0.001, 0.01, 0.1, and 1 M. In the control group bacteria were present (10.0 × 10^5^ CFU/mL). After treatment with 1 M H-002119-00-001, the number of bacteria was estimated as 1.1 × 10^2^ CFU/mL. The *H. pylori* eradication rate of H-002119-00-001 at a concentration of 1 M was 99.9% (Figure 2A,B). However, the other doses of H-002119-00-001 did not show the significant antimicrobial efficacy against H. pylori KCTC12083 (Appendix A).

### 3.2. CLO Test and qPCR for H. Pylori in the Gastric Mucosa

The CLO test was performed on the extracted gastric tissue to determine the cure rate. When H-002119-00-001 was administered to the mice once daily for 2 weeks, the treatment rate for Group 6 was 40.0%, for Group 7 it was 60.0%, and for Group 8 it was 70.0% (Table 1). When H-002119-00-001 was administered to the mice once daily for 4 weeks, the treatment rate for Group 6 was 60.0%, for Group 7 it was 55.6%, and for Group 8 it was 80.0% (Table 1). *H. pylori* treatment rate of H-002119-00-001 increased dose-dependently. qPCR for *H. pylori* was conducted to evaluate the therapeutic effect of H-002119-00-001 in *H. pylori*-infected mice. When H-002119-00-001 was administered to the mice once daily for 2 weeks, there was a significant decrease in *H. pylori* levels in Group 8 (57.1%) compared to that in Group 3 (Figure 3A). When H-002119-00-001 was administered to the mice once daily for 4 weeks, the *H. pylori* levels in Group 8 showed a statistically significant decrease (58.6%) compared to that in Group 3 (Figure 3B). 

### 3.3. Gastric Histopathological Analysis

To determine whether the anti-*H. pylori* effect of H-002119-00-001 influences gastric mucosal inflammation, pathological changes in the gastric tissue were evaluated and scored. To measure the total pathological score, the damage of the surface epithelium, inflammatory cell infiltration, and submucosal edema were observed and calculated. When H-002119-00-001 was administered to the mice once daily for 2 weeks, H-002119-00-001 significantly decreased the inflammation in gastric tissues (Figure 4A, Table 2). When H-002119-00-001 was administered to the mice once daily for 4 weeks, H-002119-00-001 also significantly decreased the inflammation in gastric tissues (Figure 4B, Table 2).

### 3.4. Gastric Tissue Immunostaining Assay 

We also evaluated macrophages in the gastric tissue to determine the anti-*H. pylori* effects of H-002119-00-001 using immunohistochemistry (IHC). IHC revealed a marked decrease in intramucosal F4/80-positive macrophages in H-002119-00-001-treated gastric tissue, both in the 2-week and 4-week treatments (Figure 5, Table 3). 

## 4. Discussion

This study was performed to evaluate the antimicrobial efficacy of H-002119-00-001 in a mouse model infected with *H. pylori* compared to that of antibiotic treatment. Moreover, in this study, we focused on identifying new therapeutic agents from natural plants for the eradication of *H. pylori*—complementary to drugs derived from synthetic sources—that have few side effects and low toxicity. H-002119-00-001 used in this study is a β-caryophyllene, a natural compound found in essential oils of many plants such as cloves, cannabis sativa, rosemary, and hops [28,29]. Essential oils have been shown to inhibit *H. pylori* in in vitro [30,31] and in vivo studies [32]. However, β-caryophyllene from essential oils has not been reported for the treatment of *H. pylori* infection. β-caryophyllene has been known to have anti-inflammatory effects and has showed efficacy with respect to the management of degenerative brain diseases such as depression, cerebral ischemic injury, Alzheimer’s disease, and epilepsy [33,34,35,36,37]. In addition, β-caryophyllene is effective against lymphoma and neuroblastoma because of its anti-inflammatory effects [38]. Therefore, we thought that the anti-inflammatory, antioxidant, and antibacterial activity of β-caryophyllene might be effective in inhibiting *H. pylori* infection and inflammation in the gastric mucosa. Thus, in this study, we investigated the bactericidal effect of H-002119-00-001 and compared it with the efficacy of antibiotic therapy.

First, we determined the antimicrobial efficacy of H-002119-00-001 using *H. pylori* KCTC12083 to evaluate the conventional in vitro bactericidal activities. Our result revealed that the *H. pylori* KCTC 12083 eradication rate of H-002119-00-001 at a concentration of 1 M was 99.9%. Based on this positive result, we conducted a preclinical study to compare these effects with those of the existing antibiotic therapy for eradication of *H. pylori*. Antibiotic therapy has high efficacy, but it is associated with the development of antibiotic resistance and numerous adverse effects such as drug allergy and gastrointestinal symptoms. The frequent occurrence of adverse effects of antibiotics can lead to reduced compliance of patients. Therefore, because of the increasing rate of antibiotic resistance and adverse effects, it is important to search for new therapeutic agents against *H. pylori* infection. Recently, natural products from plants have received attention for the discovery of new therapeutic agents for the treatment of *H. pylori*. In this preclinical study, no severe adverse effects were observed, and treatment with H-002119-00-001 did not have any significant effect on body weight in mice.

The CLO test and *H. pylori* PCR in the gastric mucosa after treatment showed that the treatment rate increased as the dose of H-002119-00-001 increased. Group 8 had the highest cure rate (70% for 2-week treatment and 80% for 4-week treatment) among the treatment groups. Furthermore, the CLO score for Group 8 was the lowest among the treatment groups (0.50 ± 0.85 for 2-week treatment and 0.50 ± 1.08 for 4-week treatment). The anti-*H. pylori* effect of H-002119-00-001 was found to be dose-dependent. 

Interestingly, H-002119-00-001-treated mice had significantly reduced levels of *H. pylori*-induced inflammation compared to that in *H. pylori*-infected but untreated mice. *H. pylori*-induced chronic inflammation contributes to disease pathogenesis [39]. Therefore, a reduction in inflammation, as observed here, is expected to reduce *H. pylori*-induced disease progression.

Our current study has a limitation. The antibiotic treatment used for comparison of the efficacy of H-002119-00-001 against *H. pylori* was composed of metronidazole, clarithromycin, and omeprazole. These antibiotics could be candidates for the first-line treatment of *H. pylori*. However, the current STT is a combination of clarithromycin, amoxicillin, and PPI, but we did not compare the efficacy of H-002119-00-001 with that of STT, the current worldwide standard for *H. pylori* treatment. In addition, the mechanism underlying the inhibitory effect of H-002119-00-001 with respect to *H. pylori* growth is still unknown. Furthermore, it is unclear whether the anti-inflammatory effect after H-002119-00-001 treatment was caused by the decreased *H. pylori* number or directly affected H-002119-00-001 to host immune cells. Therefore, further studies about the mechanism underlying the anti-inflammatory effect of the H-002119-00-001 treatment are needed. We evaluated the macrophages in the gastric tissue using immunohistochemistry (IHC). In addition, it is also important to note the neutrophils infiltration in *H. pylori*-infected gastric mucosa. However, we did not evaluate the neutrophil infiltrations by IHC analysis.

In conclusion, we demonstrated a potent antimicrobial activity of H-002119-00-001 against *H. pylori*. H-002119-00-001 treatment decreased the bacterial burden in vitro as well as in vivo. In addition to its direct killing effect, H-002119-00-001 inhibited the inflammation of the gastric mucosa. Overall, our data suggest that H-002119-00-001 is a promising candidate as an anti-*H. pylori* agent.

## Figures and Tables

**Figure 1 nutrients-12-01000-f001:**
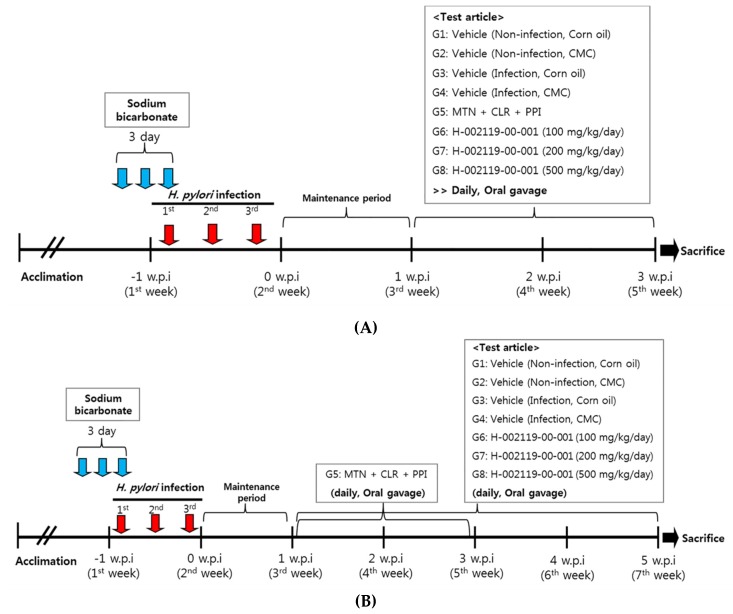
Design of animal experiments. Efficacy of H-002119-00-001 against *H. pylori* in vivo. This figure depicts the study protocol, including *H. pylori* inoculation and development of infection in C57BL/6 mice, followed by the treatments. (**A**) Two week treatment with H-002119-00-001, (**B**) Four week treatment with H-002119-00-001

**Figure 2 nutrients-12-01000-f002:**
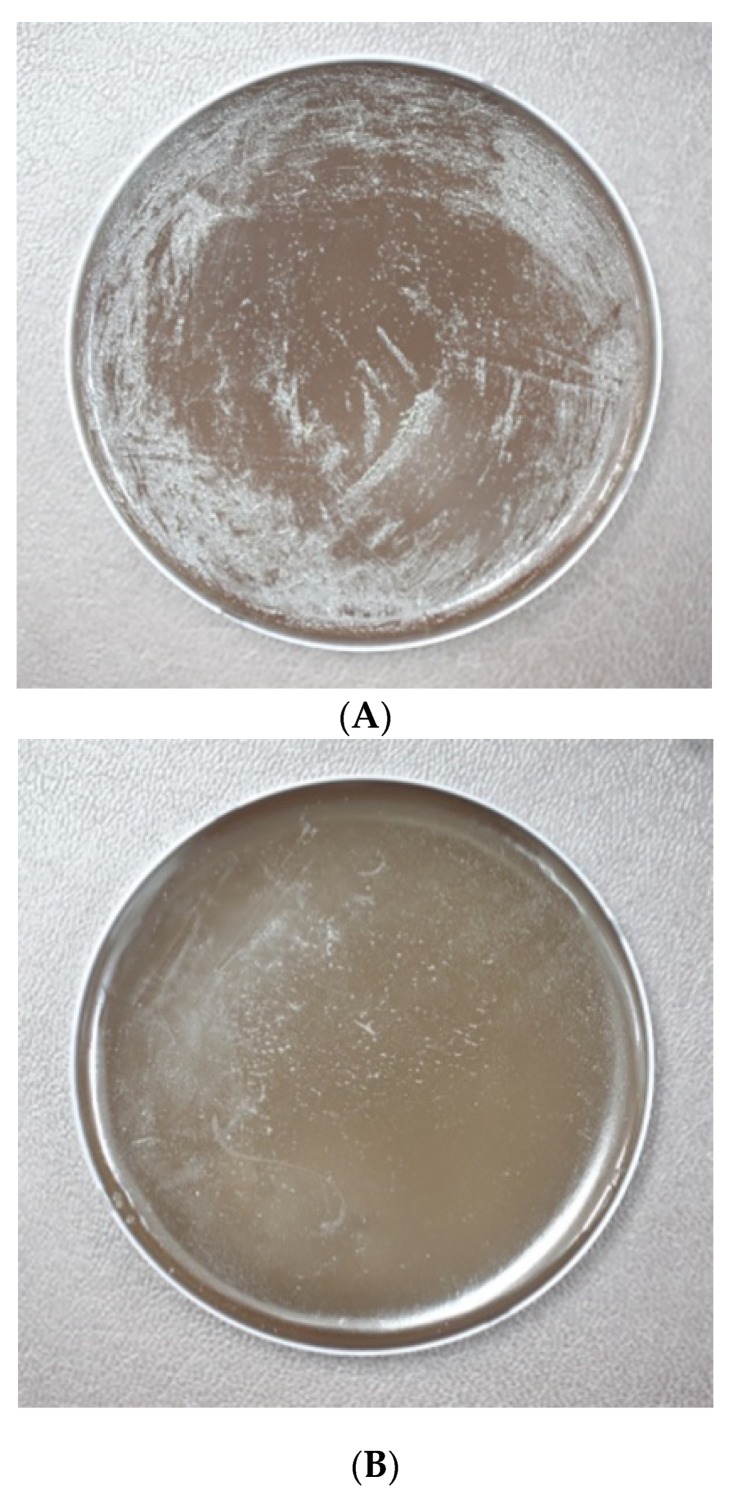
Anti-*H. pylori* efficacy of H-002119-00-001 in vitro. (**A**) The no-treatment group served as the control group. (**B**) After treatment with 1 M H-002119-00-001, the number of bacteria decreased from 10.0 × 10^5^ to 1.1 × 10^2^ CFU/mL.

**Figure 3 nutrients-12-01000-f003:**
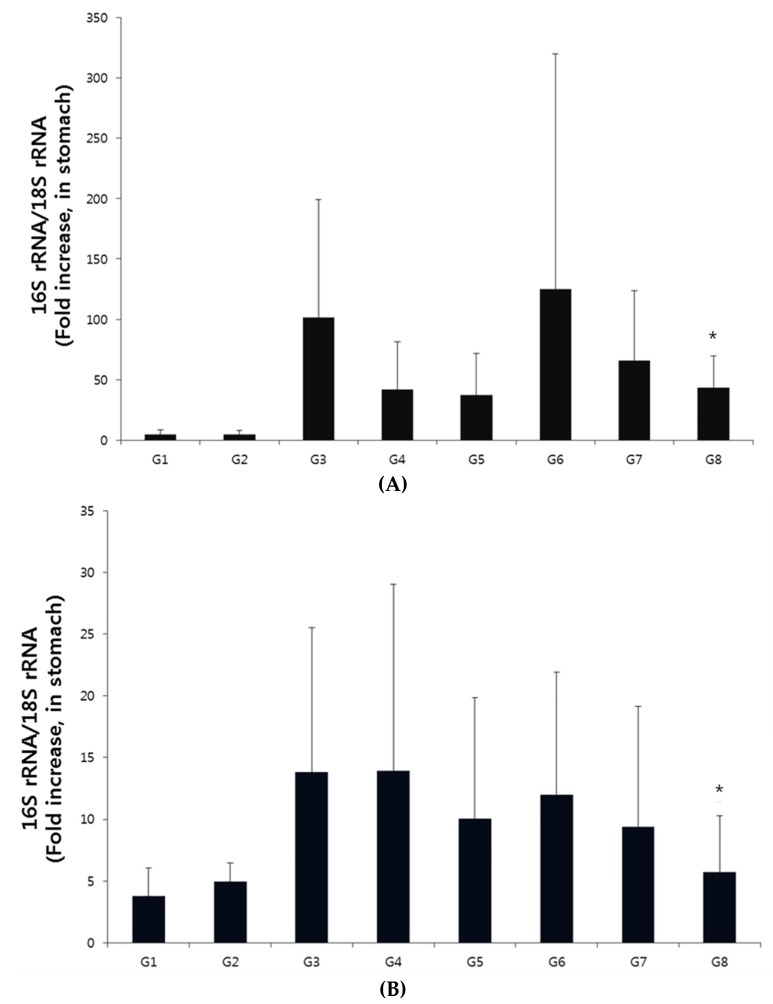
Effect of H-002119-00-001 on the qPCR score in gastric tissue. (**A**) Two-week treatment with H-002119-00-001, * Significantly different from Group III (*p* < 0.05). (**B**) Four-week treatment with H-002119-00-001. * Significantly different from Group III (*p* < 0.05).

**Figure 4 nutrients-12-01000-f004:**
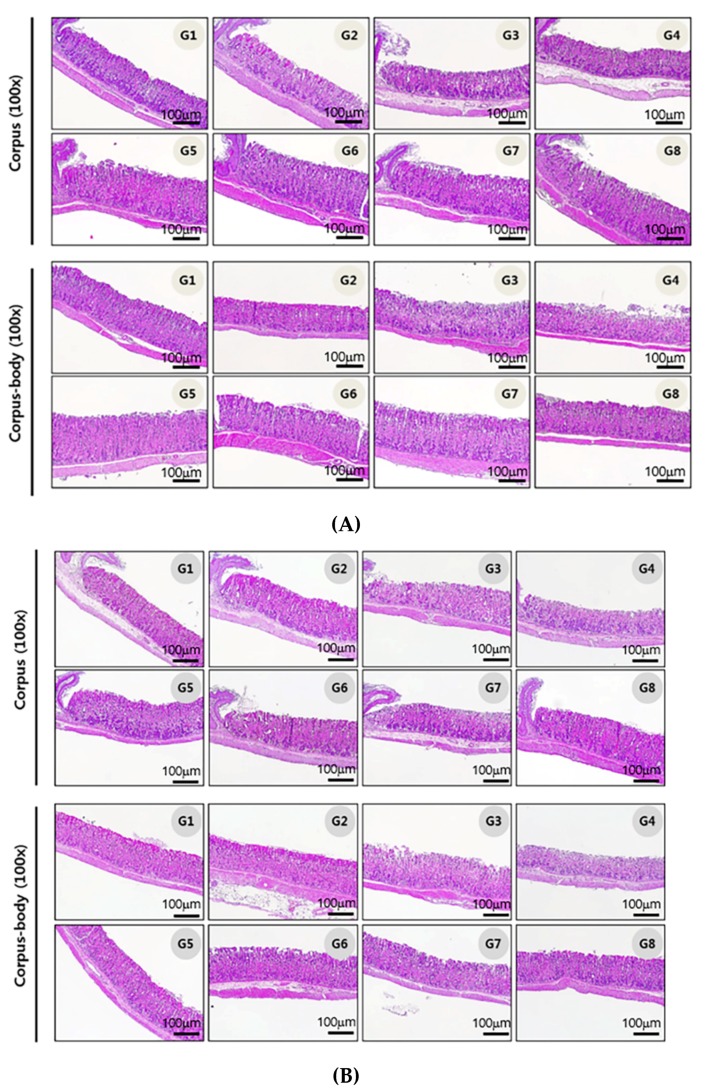
Effect of H-002119-00-001 on *H. pylori* infection-induced inflammatory cell infiltration. (**A**) Two-week treatment with H-002119-00-001, (**B**) Four-week treatment with H-002119-00-001.

**Figure 5 nutrients-12-01000-f005:**
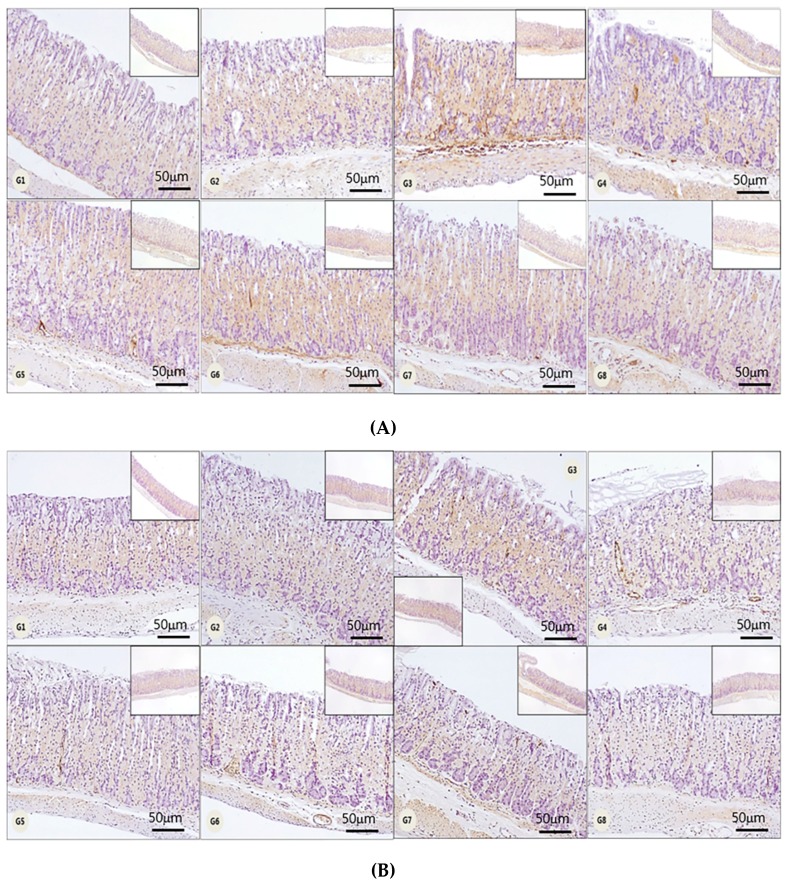
Effect of H-002119-00-001 on *H. pylori* infection-induced immune cell infiltration. (**A**) Two-week treatment with H-002119-00-001, (**B**) Four-week treatment with H-002119-00-001.

**Table 1 nutrients-12-01000-t001:** Results of the Campylobacter-like organism (CLO) test with gastric mucosa after treatment.

Group	*H. pylori* Infection	Treatment	2-Week Treatment			4-Week Treatment		
			n	Percentage of Animals Negative by CLO Test (%)	CLO Score	n	Percentage of Animals Negative by CLO Test (%)	CLO Score
I	no	corn oil	10	100	0.00 ± 0.00	10	100	0.00 ± 0.00
II	no	0.5% CMC	9	100	0.00 ± 0.00	9	100	0.00 ± 0.00
III	yes	corn oil	10	0	3.00 ± 0.00	10	30	2.10 ± 1.45
IV	yes	0.5% CMC	9	0	2.78 ± 0.44	10	40	1.70 ± 1.49
V	yes	MTN + CLR + PPI	10	80	0.20 ± 0.42	10	70	0.90 ± 1.45
VI	yes	100 mg/kg H-002119-00-001	10	40	1.30 ± 1.25 *	10	60	1.20 ± 1.55
VII	yes	200 mg/kg H-002119-00-001	10	60	0.50 ± 0.71 *	9	56	1.11 ± 1.45
VIII	yes	500 mg/kg H-002119-00-001	10	70	0.50 ± 0.85 *	10	80	0.50 ± 1.08 ^*^

* Significantly different from Group III (*p* < 0.05).

**Table 2 nutrients-12-01000-t002:** Results of histopathological examination in gastric tissue.

Group	2-Week Treatment					4-Week Treatment				
	n	Damage of the Surface Epithelium	Inflammatory Cell Infiltration	Submucosal Edema	Total Score	n	Damage of the Surface Epithelium	Inflammatory Cell Infiltration	Submucosal Edema	Total Score
I	10	0.25 ± 0.35	0.25 ± 0.26	0.10 ± 0.21	0.60 ± 0.61	10	0.05 ± 0.16	0.15 ± 0.24	0.15 ± 0.24	0.35 ± 0.47
II	9	0.28 ± 0.26	0.28 ± 0.26	0.11 ± 0.22	0.67 ± 0.50	9	0.11 ± 0.22	0.22 ± 0.26	0.17 ± 0.25	0.50 ± 0.43
III	10	0.65 ± 0.34	0.50 ± 0.33	0.55 ± 0.37	1.70 ± 0.67	10	0.65 ± 0.24	0.45 ± 0.37	0.45 ± 0.28	1.55 ± 0.55
IV	9	0.61 ± 0.33	0.56 ± 0.30	0.44 ± 0.46	1.61 ± 0.74	10	0.60 ± 0.32	0.55 ± 0.37	0.50 ± 0.41	1.65 ± 0.88
V	10	0.30 ± 0.42	0.55 ± 0.44	0.15 ± 0.24	1.00 ± 0.78	10	0.35 ± 0.34	0.40 ± 0.39	0.35 ± 0.34	1.10 ± 0.77
VI	10	0.30 ± 0.35 *	0.40 ± 0.32	0.25 ± 0.35 *	0.95 ± 0.60 *	10	0.40 ± 0.21 *	0.40 ± 0.32	0.35 ± 0.34	1.15 ± 0.67
VII	10	0.35 ± 0.34 *	0.25 ± 0.35	0.25 ± 0.35 *	0.85 ± 0.88 *	9	0.39 ± 0.42	0.44 ± 0.39	0.39 ± 0.33	1.22 ± 1.00
VIII	10	0.25 ± 0.26 *	0.25 ± 0.26 *	0.15 ± 0.34 *	0.65 ± 0.63 *	10	0.25 ± 0.26 *	0.35 ± 0.24	0.25 ± 0.42	0.85 ± 0.53 *

* Significantly different from Group III (*p* < 0.05).

**Table 3 nutrients-12-01000-t003:** Results of immunohistochemical assessments in gastric tissue.

Group	2-Week Treatment		4-Week Treatment	
	n	F4/80 Score	n	F4/80 Score
I	10	0.10 ± 0.21	10	0.10 ± 0.21
II	9	0.06 ± 0.17	9	0.15 ± 0.24
III	10	0.65 ± 0.34	10	0.58 ± 0.20
IV	9	0.44 ± 0.30	10	0.35 ± 0.34
V	10	0.55 ± 0.44	10	0.25 ± 0.35
VI	10	0.35 ± 0.24 *	10	0.35 ± 0.41
VII	10	0.20 ± 0.26 *	9	0.22 ± 0.26
VIII	10	0.25 ± 0.26 *	10	0.20 ± 0.26 *

* Significantly different from Group III (*p* < 0.05).

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
