# Peer review of "Effect of β-caryophyllene from Cloves Extract on *Helicobacter pylori* Eradication in Mouse Model"

_nutrients, 2020, doi:10.3390/nu12041000_

Round 1

Reviewer 1 Report

In my opinion the authors should describe more H-002119-00-001, so readers would like to know more about compounds of H-002119-00-001. please exmplain more about β-caryophyllene that only: is a natural
bicyclic sesquiterpene that is present in a wide range of plant species such as cloves, basil, and cinnamon. It has known to have anti-inflammatory, antioxidant, anticancer, and antimicrobial effects. H-002119-00-001 is obtained from β-caryophyllene extract.  Where we can find β-caryophyllene, how many amount we have in the different plants and also how do you prepare the extract.

In references you should put also:

https://www.ncbi.nlm.nih.gov/pubmed/29075827

Appl Microbiol Biotechnol. 2018 Jan;102(1):1-7. doi: 10.1007/s00253-017-8535-7. Epub 2017 Oct 26.

Helicobacter pylori treatment: antibiotics or probiotics.

Goderska K, Agudo Pena S, Alarcon T.

Author Response

Reviewer’s comments

Reviewer 1

In my opinion the authors should describe more H-002119-00-001, so readers would like to know more about compounds of H-002119-00-001. please explain more about β-caryophyllene that only: is a natural bicyclic sesquiterpene that is present in a wide range of plant species such as cloves, basil, and cinnamon. It has known to have anti-inflammatory, antioxidant, anticancer, and antimicrobial effects. H-002119-00-001 is obtained from β-caryophyllene extract. Where we can find β-caryophyllene, how many amount we have in the different plants and also how do you prepare the extract.

Answer

We thank you for your valuable comment. As your comment, we added sentences about β-caryophyllene into the introduction like below (2page, line 48 – 55).

==================================================================

β-caryophyllene is a natural bicyclic sesquiterpene that is present in a wide range of plant species such as cloves, basil, cinnamon leaves, and copaiba balsam. [17] It ((1R,4E,9S)-4,11,11-trimethyl-8-methylidenebicyclo[7.2.0]undec-4-ene) is a major constituent of many essential oils obtained from a number of plant species such as Syzygium (13%), Betula (30%) and Strobilanthes (7%) species. [18-20] It has known to have anti-inflammatory, antioxidant, anticancer, and antimicrobial effects. In this study, we used the β-caryophyllene from cloves which are the aromatic flower buds of a tree, Syzygium aromaticum in the family Myrtaceae. For the preparation of β-caryophyllene, steam distillation of dried flower’s buds was performed and then the essential oil was obtained. After fractional distillation and concentration of essential oil, β-caryophyllene was prepared.

==================================================================

  1. In references you should put also:https://www.ncbi.nlm.nih.gov/pubmed/29075827 Appl Microbiol Biotechnol. 2018 Jan;102(1):1-7. doi: 10.1007/s00253-017-8535-7. Epub 2017 Oct 26. Helicobacter pylori treatment: antibiotics or probiotics. Goderska K, Agudo Pena S, Alarcon T.

Answer

We thank you for your valuable comment. As your comment, we added the references like below (1page, line 43).

==================================================================

Therefore, there is an increasing demand for safe and effective non-antibiotic compounds that inhibit the growth of H. pylori. [11-13]

==================================================================

Reviewer 2 Report

The present study evaluated the antibacterial effect of H-002119-00-001 derived from β-caryophyllene against Helicobacter pylori in vitro and in vivo examination. As a result, it is shown that H. pylori bactericidal effect of H-002119-00-001 at a concentration of 1 M was 99.9%. Additionally, from in vivo mice-H. pylori infection models analysis, it is shown that the H. pylori treatment rate of H-002119-00-001 increased dose-dependently and H-002119-00-001 significantly decreased the inflammation in gastric tissues.

1) The naming of "H-002119-00-001" is very confusing and needs to be revised. It is better to add the structural formula of H-002119-00-001.

2) In in vitro analysis, Antibacterial activity against H. pylori was evaluated with 1M H-002119-00-001. 1M is a very high concentration. It is necessary to evaluate the capacity dependency of H-002119-00-001 on the antibacterial effect.

3) In in vivo analysis, From Fig. 4 data, it seems to be hardly detected infiltration of inflammatory cells.  It is necessary that H. pylori infection is assessed by colony-forming units (CFU) assay.

4) In Fig. 5, it was assessed the infiltration of macrophages by using F4/80 immunostaining (IHC). However, in H. pylori-infected gastric mucosa, it is also important that neutrophils infiltration. So, the author needs to evaluate the neutrophil infiltrations by IHC analysis.

5) The author concluded that H-002119-00-001 inhibited the inflammation of the gastric mucosa-infected with H. pylori. However, it is not clear whether this anti-inflammatory effect was caused by the decreased H. pylori number or directly affected H-002119-00-001 to host immune cells. Additional discussions against this phenomenon need to describe.

Author Response

Reviewer’s comments

Reviewer 2

The present study evaluated the antibacterial effect of H-002119-00-001 derived from β-caryophyllene against Helicobacter pylori in vitro and in vivo examination. As a result, it is shown that H. pylori bactericidal effect of H-002119-00-001 at a concentration of 1 M was 99.9%. Additionally, from in vivo mice-H. pylori infection models analysis, it is shown that the H. pylori treatment rate of H-002119-00-001 increased dose-dependently and H-002119-00-001 significantly decreased the inflammation in gastric tissues.

  1. The naming of "H-002119-00-001" is very confusing and needs to be revised. It is better to add the structural formula of H-002119-00-001.

Answer

We thank you for your valuable comments. H-002119-00-001 is obtained from β-caryophyllene extract. We agree with your opinion that H-002119-00-001 is confusing and needs to be revised. Therefore, we changed it to β-caryophyllene in title and added sentences like below (2page, line 48 – 50).

==================================================================

Title : Effect of H-002119-00-001 β-caryophyllene from cloves extract on Helicobacter pylori eradication in mouse model

==================================================================

β-caryophyllene is a natural bicyclic sesquiterpene that is present in a wide range of plant species such as cloves, basil, cinnamon leaves, and copaiba balsam. [17] It ((1R,4E,9S)-4,11,11-trimethyl-8-methylidenebicyclo[7.2.0]undec-4-ene) is a major constituent of many essential oils obtained from a number of plant species such as Syzygium (13%), Betula (30%) and Strobilanthes (7%) species. [18-20] It has known to have anti-inflammatory, antioxidant, anticancer, and antimicrobial effects.

==================================================================

  1. In in vitro analysis, antibacterial activity against H. pylori was evaluated with 1M H-002119-00-001. 1M is a very high concentration. It is necessary to evaluate the capacity dependency of H-002119-00-001 on the antibacterial effect.

Answer

We thank you for your valuable comments. We evaluated the capacity dependency of H-002119-00-001 in vitro (0.001M, 0.01M, and 0.1M, respectively). However, the other doses except 1M H-002119-00-001 showed over 3.0 x 102 CFU/mL after H-002119-00-001 treatment. As your comments, we added this results into the results section like below (5page, line 156 – 157, line 160 – 161).

==================================================================

The antimicrobial efficacy of H-002119-00-001 against H. pylori KCTC12083 was evaluated at a dose of 0.001M, 0.01M, 0.1M and 1M. In the control group bacteria were present (10.0 × 105 CFU/mL). After treatment with 1 M H-002119-00-001, the number of bacteria was estimated as 1.1 × 102 CFU/mL. The H. pylori eradication rate of H-002119-00-001 at a concentration of 1 M was 99.9% (Figure 2A, B). However, the other doses of H-002119-00-001 did not show the significant antimicrobial efficacy against H. pylori KCTC12083 (Supplementary Figure 1).

==================================================================

Supplementary Figure 1. In vitro effect of H-002119-00-001 on bacterial colonization (A) After treatment with 0.1 M H-002119-00-001, the number of bacteria showed over 3.0 x 102 CFU/mL. (B) After treatment with 0.01 M H-002119-00-001, the number of bacteria showed over 3.0 x 102 CFU/mL. (C) After treatment with 0.001 M H-002119-00-001, the number of bacteria showed over 3.0 x 102 CFU/mL.

(A)

(B)

(C)

==================================================================

  1. In in vivo analysis, From Fig. 4 data, it seems to be hardly detected infiltration of inflammatory cells. It is necessary that H. pylori infection is assessed by colony-forming units (CFU) assay.

Answer

We thank you for your valuable comments. We evaluated the histotopathological scores including inflammatory cell infiltration by two different researchers to increase objectivity. As your comments, it would be better to assess H. pylori infection by colony-forming units (CFU) assay. However, we evaluated the H. pylori infection by H. pylori antibody (IgG), campylobacter-like organisms (CLO) test, and real-time polymerase chain reaction (qPCR).

  1. In Fig. 5, it was assessed the infiltration of macrophages by using F4/80 immunostaining (IHC). However, in H. pylori-infected gastric mucosa, it is also important that neutrophils infiltration. So, the author needs to evaluate the neutrophil infiltrations by IHC analysis (.

Answer

We thank you for your valuable comments. As we mentioned before, we evaluated the infiltration of inflammatory cells including neutrophils, etc. As shown in below figure, the pathologist evaluated the infiltration of inflammatory cells and then analyzed the other tissues by same methods. However, we did not evaluate the neutrophil infiltrations by IHC analysis. As your comments, we added sentences into the limitation like below (12page, line 273 – 13 page line 276).

Gastric mucosa of H. pylori-infected mice displayed inflammation with surface epithelium loss (green arrow), inflammatory cell infiltration (black arrow) and submucosal edema (black arrowhead).

==================================================================

In addition, the mechanism underlying the inhibitory effect of H-002119-00-001 with respect to H. pylori growth is still unknown. We evaluated the macrophages in the gastric tissue using immunohistochemistry (IHC). And, it is also important the neutrophils infiltration in H. pylori-infected gastric mucosa. However, we did not evaluate the neutrophil infiltrations by IHC analysis.

==================================================================

  1. The author concluded that H-002119-00-001 inhibited the inflammation of the gastric mucosa-infected with H. pylori. However, it is not clear whether this anti-inflammatory effect was caused by the decreased H. pylori number or directly affected H-002119-00-001 to host immune cells. Additional discussions against this phenomenon need to describe.

Answer

We thank you for your valuable comment. We agree with your comments that it is unclear whether anti-inflammatory effect after H-002119-00-001 treatment was caused by the decreased H. pylori number or directly affected H-002119-00-001 to host immune cells. However, the β-caryophyllene has been known to have anti-inflammatory effects, and has showed efficacy with respect to the management of various diseases. Therefore, we thought that β-caryophyllene had the direct killing effect against H. pylori and also inhibited the inflammation of the gastric mucosa. However, further studies about exact mechanism are needed as your comment. Therefore, we added sentences into the discussion like below (12page line 270 – 273).

==================================================================

In addition, the mechanism underlying the inhibitory effect of H-002119-00-001 with respect to H. pylori growth is still unknown. And, it is unclear whether anti-inflammatory effect after H-002119-00-001 treatment was caused by the decreased H. pylori number or directly affected H-002119-00-001 to host immune cells. Therefore, further studies about the mechanism underlying anti-inflammatory effect of H-002119-00-001 treatment are needed.

==================================================================

Round 2

Reviewer 2 Report

Thank you for revision of your manuscript.
In Figs 4 and 5, Scale bar is needed.
Nothing further comments.

Author Response

Reviewer’s comments

In Figs 4 and 5, Scale bar is needed.

Answer

We thank you for your valuable comment. As your comment, we amended Figure 4 and 5 like below. ==================================================================

Figure 4. Effect of H-002119-00-001 on H. pylori infection-induced inflammatory cell infiltration. A. Two-week treatment with H-002119-00-001, B. Four-week treatment with H-002119-00-001.

A

B

Figure 5. Effect of H-002119-00-001 on H. pylori infection-induced immune cell infiltration. A. Two-week treatment with H-002119-00-001, B. Four-week treatment with H-002119-00-001.

A

B
